# The Future Is Hybrid: How Organisations Are Designing and Supporting Sustainable Hybrid Work Models in Post-Pandemic Australia

John Hopkins *[ID] and Anne Bardoel [ID]

Department of Management and Marketing, Swinburne University of Technology, Hawthorn, VIC 3122, Australia
* Correspondence: jlhopkins@swin.edu.au

**Abstract:** Hybrid work models have rapidly become the most common work arrangement for many knowledge workers, affording them with improved work–life balance and greater levels of job satisfaction, but little research has been conducted to identify the different hybrid work models that are emerging, and the appropriate supports needed to drive sustainable improvement. This paper utilises primary data from a series of semi-structured interviews with senior Australian human resource (HR) managers, to identify a range of different approaches to hybrid work design, applying the Conservation of Resources (COR) theory. Analyses of these findings have resulted in five key contributions: one being the identification of the most popular current hybrid work arrangements; the second being the key supporting pillars that are required to support successful hybrid work; the third identifies the infrastructure required to support these pillars; the fourth being a theoretic contribution that extends the existing academic literature in this field; and with the final contribution being an interpretation of the findings via COR theory. These contributions have significant implications for both scholars and human resource professionals, as organisations and academics strive to learn from the recent period of turbulence and develop sustainable improvements in performance and working conditions (SDG8), with improved support for employee health and wellbeing (SDG3), and gender equality (SDG5).

**Keywords:** hybrid work models; flexible work arrangements; work–life balance; human resource management; health and wellbeing; hybrid work culture; future of work; telework

## 1. Introduction

In the aftermath of the COVID-19 pandemic, the most common work arrangement for knowledge workers is now some form of a hybrid model, a type of 'flexible working where an employee splits their time between the workplace and remote working' [1]. Arrangements such as these offer workers a greater level of control, over where (location) and when (timing) their work tasks are performed, leading to a potential for improvements in flexibility, autonomy, and work–life balance [2,3].

In June 2022, Gallup surveyed a nationally representative sample of 8090 remote-capable employees in the US and found that 60% now prefer hybrid work options [4]. Similarly, a July 2022 pulse survey from Future Forum claimed that hybrid is already the dominant business model for 49% of 10,646 knowledge workers around the globe [5], and an April 2022 study of 1421 Australian knowledge workers found that 54% were now following some type of hybrid work model, too, with 56% indicating this was also the ideal work arrangement for them [6].

Recent studies indicate that hybrid working offers many positive outcomes for both employers and employees, with reports linking it to a 35% reduction in attrition [7], without any adverse impact on performance or a worker's promotion opportunities, and significantly improved employee experience scores [5]. To underline this, global brands such as

Apple, Meta, Citi Standard Chartered Bank, HSBC, Volkswagen AG, and Bupa have all spoken publicly about how they are embracing hybrid working arrangements [8,9].

The United Nations (UN) Sustainable Development Goals (SDGs), a set of 17 goals designed as a "universal call to action to end poverty, protect the planet, and ensure that by 2030 all people enjoy peace and prosperity" [10], include commitments to decent work (SDG8), gender equality (SDG5), and good health and wellbeing (SDG3). As identified by [11], flexible work arrangements (FWAs) such as telework and hybrid working offer many strengths and opportunities for supporting SDGs.

Despite this, some negative impacts have also been reported. The lack of job visibility and predictability, social isolation and level of uncertainty associated with hybrid working, have been found to be a source of stress for some workers [12,13]. Yelp CEO Jeremy Stoppelman famously called hybrid offices 'the hell of half measures' and 'the worst of both worlds', saying his organisation would focus on moving fully remote instead [14].

Prior to the pandemic, FWAs such as telework were only available to a relatively small section of the workforce [15], to a point where they were even described as an "elitist phenomenon" [16]. However, COVID-19 quickly changed that and employees across the globe are now, perhaps understandably, reluctant to give up their newly-acquired levels of work flexibility and the improved quality of life it offers them [17], to the point where many are now believed to value it even more than a pay rise [18].

The transition to working from home (WFH) during the pandemic was made possible by the widespread availability and maturity of modern information and communications technology (ICT), and has resulted in a dramatic shift in terms of how organisations and employees think about work and the role of the physical office, being described as a "major change in the labour market that has occurred at unprecedented speed" [19],

This emergence of hybrid working as the 'new normal' for knowledge workers makes it a vital area for academic investigation, and this research aims to better understand the nature of this phenomenon and how organisations might successfully leverage its potential, by providing the appropriate technologies, infrastructure, and processes to support it.

## 2. Theoretical Background

### 2.1. Key Terms and Definitions

In this section, we revisit and discuss a number of key terms and definitions associated with this field of study, in order to position the emerging phenomenon of hybrid work within the existing body of academic literature.

### 2.1.1. Hybrid Work

Hybrid work is a relatively new term, gaining popularity during the pandemic to define a working arrangement where an employee divides their time between working at a traditional workplace and working remotely (typically at home, or from 'third places' such as a coworking space, library, or local café, etc.), which attempts to combine the best parts of both telework and office-based work [1,11]. It describes an employee's ability to have some level of autonomy and flexibility over the *location* where they perform their work tasks, and builds upon Halford's research into *hybrid workplaces*, which investigated the use of ICT to "maintain workloads and relationships across both domestic and organisational spaces. (where) individuals work at home and engage in embodied organisational spaces" almost 20 years ago [20].

It is important to note at this point, that contemporary thinking regarding the topic of 'hybrid work' is very different to that of 'hybrid work characteristics', which was a term used in psychology to describe "work characteristics which are not fully captured within any one of the three domains (task, social, or contextual) but possess features from more than one domain" [21].

### 2.1.2. Telecommuting and Telework

Whilst hybrid work might be a relatively new term, studies into the concept of this kind of work behaviour can be traced back as far as the 1970s, and the work of Nilles [22]. Nilles [23] first introduced the concept of 'telecommuting', to define "an arrangement between employer and employee that allows work to be performed outside of a usual place of work on a regular basis . . . by harnessing ICT to reproduce significant aspects of the centralized work environment" [24]. Over time the term 'telecommuting' gradually became 'telework', but whilst the concept might date back almost 50 years, the practice of telework was not widely viable until much later, with the emergence of personal home computers and widespread internet connectivity in the 1990s [22,25]. Even then, once the technology to support it became available, the practice of teleworking remained relatively limited until the pandemic [16].

### 2.1.3. Third Places

The concept of 'third places' is attributed to sociologist Ray Oldenburg, who introduced it as a term to describe the places where people spend time between home and work—"locations where we exchange ideas, have a good time, and build relationships" [26]. In urban planning, third places are viewed as an important ingredient in community building, and for stabilizing neighbourhoods. They are defined as social surroundings, separate from the two usual social environments of home—"the first place", and work—"the second place" [27,28]. Examples of these third places might include cafes, libraries, bookstores, restaurants, or parks.

In the post-covid era, the concept of third places is starting to gather a lot of attention and momentum, but instead of them being regarded as places that are formally separate to work, they are increasingly being identified as alternative locations where we can work—places that are perhaps less formal than an office, but less isolating as working from home on our own.

### 2.1.4. Working from Home

Bloom et al. [29] define working from home as "also called telecommuting or telework", but the authors disagree with this for the reason that working from home is, by definition, limited to just the home. Telework is not limited to the home, telework can be conducted from pretty much anywhere, so working from home is a form of telework but not the exact same thing.

Olsone and Primps [30] also confine telework to the home, by saying it "refers to the substitution of telecommunications technology for physical travel to a central work location; it usually implies that the person is working in the home". However, the ubiquitous availability of wi-fi networks and cloud computing, enabling knowledge workers to perform their roles from public libraries, cafes, beaches, and public parks was difficult to predict in 1984. Therefore, the authors reiterate that whilst the definition of teleworking has expanded over time to include a variety of non-home locations, the term *working from home* should only be used to describe home-based work activities.

### 2.1.5. Remote Work

Remote work describes the "organization and/or performance of work, whereby an employee can carry out work that could also be carried out at the employer's premises regularly out of these premises through the use of information technology" [31]. The key difference here is the inclusion of the word *regularly*, and remote workers are often regarded as spending the majority of their time, or in many cases *all* their time, working away from the employer's premises.

Work from anywhere (WFA), or anywhere working, is a form of remote work popularized by *digital nomads*, workers whose lives are completely location-independent. They use digital technologies to perform their job, whilst living a nomadic lifestyle, where they are able to combine working and travelling the world [32,33]. They typically perform

their work tasks from their temporary home or a mixture of different third places, and the number of people choosing to live this digital nomad lifestyle is increasing significantly. So much so, that many countries are introducing special post-pandemic visa and tax schemes, in an attempt to attract more digital nomads to their region [34].

Whilst all these terms describe some type of flexible work arrangement, and there are clear similarities and overlaps between each of them, it is important to emphasise the nuances and position hybrid work appropriately. For the purposes of this research, the authors have developed the following model, to illustrate the positioning of hybrid work in the existing body of academic literature (see Figure 1).

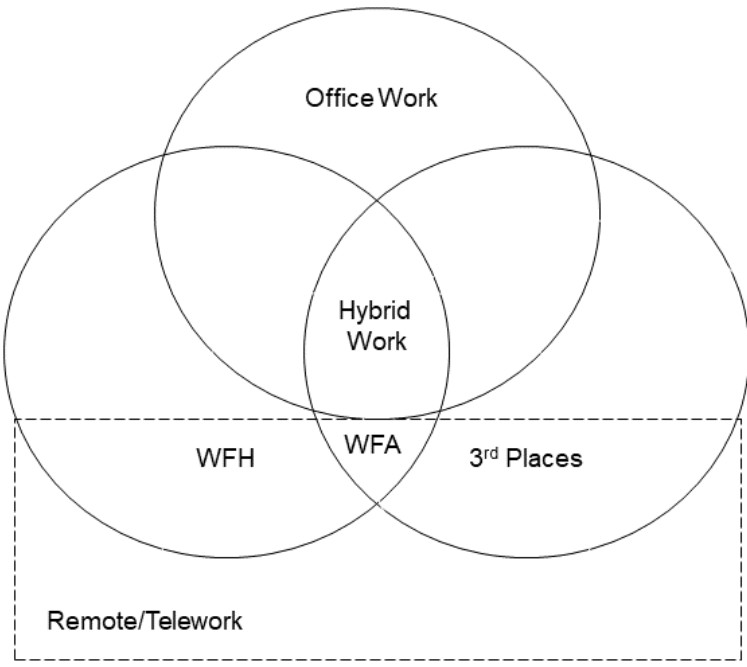

**Figure 1.** Positioning hybrid work within the existing academic literature.

### 2.2. Conservation of Resources Theory

The conservation of resources (COR) theory postulates that individuals strive to limit the loss of resources (e.g., time, cognitive, and emotional) and gain additional resources to achieve their goals. We offer an approach to understanding how hybrid work can be understood from a resource perspective [35], and believe COR theory can be used to explain the emergence of institutional logic, leading from WFH during lockdowns to hybrid work arrangements.

Whilst research into FWAs and telework practices is relatively extensive [23,36–40], work conducted prior to the pandemic was concerned with a practice which was restricted to a small minority of workers, whereas now it is something that is much more widely available and demanded. As such, for many organisations (and particularly managers), it is still a relatively new concept for which they have limited experience. Expert guidance is needed by these organisations, to help them design and implement appropriate hybrid work arrangements, and nurture high-performing hybrid teams of their own. Therefore, this redesign of our working arrangements and workplace environments will demand new and unconventional ways of thinking, and practices that will create new workplace models for the future.

Hybrid work in the current context involves a mixture of on-site work and remote work, with the latter predominantly related to work-from-home. In many cases, the pandemic context shifted thinking around remote work from the question of whether remote work was beneficial to employees and organisations to the question of the appropriate mix of remote and on-site work, which justifies the focus on hybrid work arrangements.

Conservation of resources (COR) theory is used here to develop an understanding of hybrid work generally, but mainly to understand how it altered the environment for hybrid work in the context of the pandemic. Although originally developed as a theory of stress [41], COR theory has since been used as a framework to analyse a wide variety of adverse circumstances, ranging from natural disasters to poverty, and various workplace issues [42]. The two key tenets of COR theory are:

(1)    The primacy of resource loss. The first principle of COR theory is that resource loss is disproportionately more salient than resource gain.

(2)    Resource investment. The second principle of COR theory is that people must invest resources in order to protect against resource loss, recover from losses, and gain resources [43].

Central to understanding COR theory and hybrid work is the role of time and history. If resources are threatened, then one response may be expected to counter the threat, but if resources have already been lost, then a different response may be expected. Consider a pre-pandemic study comparing individuals who did and did not telecommute. It found that the perceived work–life balance was similar across the two groups [44], and concluded that individuals had self-selected into one or the other status, with those viewing telecommuting as a resource threat presumably avoiding it.

COR theory provides valuable insights into why hybrid work arrangements, rather than a return to past practices, have eventuated under the current circumstances. First, many employees have experienced a net resource gain in terms of autonomy and reduced commuting time, so they will strive to maintain those resources according to Principle 1. Second, to the extent necessary resources were provided for work-from-home, those investments will in large measure represent a sunk cost, which cannot be recovered even if a return to full-time on-site work occurs. Third, if those resources were provided, most employees will likely maintain if not exceed prior levels of productivity, so there will be no productivity gain for a return to past practice. Fourth, to the extent employees continue to work at home, a long-term reduction in resources expended on office space may be possible. Finally, employers may not be eager to use supervisory control (and loss of employee autonomy) as an explicit justification for a return to past practice, because the language of employee productivity is considered a stronger justification for virtually any policy.

Regardless of the specific rationale for hybrid work, further resource investments are required to switch from all office, or all remote, to hybrid work. That is, tasks need to be delineated into those best performed on-site and remotely, and then those tasks need to be coordinated. The latter is complex (hence resource-intensive) because individuals may have different working times, both generally and when they are at the workplace so, for example, meetings need to be coordinated.

### 2.3. Aim of This Research

The idea of a blended/hybrid workplace is not new, but what is new is the speed of change, and the scale of availability and access. The socio-economic 'turbulence' caused by the global pandemic has unlocked a unique opportunity for changes in the way employees connect to work [45]. The COVID-19 pandemic forced the 'discovery' for both employees and employers that remote working, and now hybrid working, is both feasible and beneficial. Strong workforce demand for hybrid working requires employers to re-engage with FWAs and consider how to design jobs and workspaces for the future [46].

Therefore, the goal of this investigation is to gain a better understanding of this new hybrid work phenomenon and extend previous academic understanding of this practice, using a post-pandemic lens. The authors aim to do this by addressing the following research questions:

RQ1  What post-pandemic hybrid work models are now emerging?

RQ2  What are the key managerial considerations for successfully leveraging hybrid work arrangements?

RQ3 What role does technology play in facilitating and supporting these emerging hybrid work models?

## 3. Materials and Methods

Semi-structured interviews were chosen for this research, as they are a flexible and versatile data collection method [47], that enable an interviewer to improvise the line of questioning based on participants' responses to earlier questions [48]. The method is sufficiently structured to enable specific dimensions of research questions to be addressed while being flexible enough to accommodate additional contributions from participants [49].

### 3.1. Research Context

As part of a review of the body of published academic management research, Voss [50] found that papers featuring multi-case investigations typically include 5–16 interviews, making the researchers' target of n = 15 interviews an appropriate number for a meaningful investigation. The interviews took place between February 2021 and October 2021, before the emergence of the Delta and Omicron variants of the coronavirus, and outside of the key lockdown periods and times when government WFH restrictions were in place. A total of n = 15 semi-structured interviews were successfully conducted, with senior HR leaders representing 15 different case organisations from around Australia, n = 12 of the participants were female and n = 3 were male, and typical role titles of the participants included Head of People and Culture, Head of Human Resources (Australia), Head of Human Resources (Asia Region), and Chief People Officer (Asia and ANZ). The interviews lasted between 33.14 min and 54.46 min, with the median being 49.24 min.

Descriptors of each participating organisation are summarised in Table 1. Each participating organisation was allocated a unique identification number, based on their Australia and New Zealand Standard Industry Code (ANZSIC). N = 12 of the participating companies were from the private sector and three were from the public sector, representing n = 12 different industries in total, ranging from Agriculture, Forestry and Fishing to Information, Media and Telecoms. N = 12 were classified as large organisations (>200 employees), n = 2 were classified as medium size organisations (20–199 employees), and n = 1 was a small organisation. N = 8 of the companies interviewed had an international focus, whilst the remaining n = 4 had a national focus, or local focus (n = 3). These data suggest that, as we hoped, a wide range of contexts would be represented in the research.

**Table 1.** Details of participating organisations.

| ID | Industry (ANZSIC) | Org. Size | % Essential | Sector | Focus | Duration |
|----|-------------------|-----------|-------------|--------|-------|----------|
| A1 | A—Agriculture, Forestry and Fishing | Medium (20–199) | 80 | Private | National | 36.56 |
| B1 | B—Mining | Large (200+) | 60 | Private | International | 50.23 |
| C1 | C—Manufacturing | Large (200+) | 10 | Private | International | 54.46 |
| C2 | C—Manufacturing | Large (200+) | 25 | Private | International | 40.55 |
| H1 | H—Accommodation and Food Services | Large (200+) | 60 | Private | International | 49.54 |
| J1 | J—Information, Media and Telecoms | Medium (20–199) | 0 | Private | International | 44.58 |
| K1 | K—Financial and Insurance Services | Large (200+) | 20 | Private | National | 52.29 |
| L 1 | L—Rental Hiring and Real Estate Services | Large (200+) | 66 | Private | International | 41.51 |
| M1 | M—Professional, Scientific and Technical Services | Large (200+) | 20 | Public | National | 51.02 |
| N1 | N—Administrative and Support Services | Large (200+) | 0 | Private | International | 44.53 |
| P1 | P—Education and Training | Large (200+) | 5 | Public | International | 50.26 |
| Q1 | Q—Health Care and Social Assistance | Large (200+) | 50 | Private | Local | 33.14 |
| Q2 | Q—Health Care and Social Assistance | Large (200+) | 12.5 | Private | National | 40.45 |
| I1 | I—Transport, Postal and Warehousing | Large (200+) | 75 | Public | Local | 43.12 |
| M2 | M—Professional, Scientific and Technical Services | Small (<20) | 5 | Private | Local | 49.24 |

The proportion of essential workers the participating organisations employed, or those who had to attend their place of work as normal throughout the pandemic, ranged from 0% to 80%. Whilst many of the organisations interviewed employed both essential and non-essential workers, the focus of this research was non-essential knowledge workers, who have the flexibility to work remotely.

Of the interview participants 6.7% (n = 1) indicated their employees were returning to the office full-time, 6.7% (n = 1) said they were moving to a fully remote model, whilst the remaining 86.7% (n = 13) said they were adopting some kind of hybrid arrangement.

The interviews included a range of questions about workplace flexibility offerings, both pre- and post-pandemic, the prior existence of and any changes that were made to organisational WFH policies, what proportion of essential/frontline workers each organisation had, pre-existing and new technologies, recruitment, training and decision-making. Prior to the interviews, desktop research was conducted into the participating organisations to capture their Australian and New Zealand Standard Industrial Classification (ANZSIC) code, organisation size, etc.—the accuracy of these details was confirmed with the interview participants at the start of each interview, along with their job title and geographical region of operation. The participating organisations were selected in such a way as to ensure a wide variety of industrial and organisational contexts would be covered. However, the authors do not claim that this particular cohort provides a representative sample of all Australian organisations.

### 3.2. Data Analysis

All interviews were conducted via the Microsoft Teams video conferencing platform and automatically transcribed using the built-in live transcription tool, then later proofread for accuracy before analysis. The researchers also took field notes throughout the interviews. The online visual collaboration software Miro was also utilised, to assist with the identification of themes during the analysis phase, and to enable the identification of any key emerging themes for managing and supporting hybrid work arrangements.

Fifteen semi-structured interviews were performed in total, but one transcription file was later found to be corrupt, leaving a sample of 14 transcripts for analysis. However, field notes were still available for the lost interview, in addition to the demographic data, etc., collected prior to the interview. Author #1 performed the initial analysis of the data, undertaking a close reading of all 14 interview transcripts and 15 sets of field notes, focusing on types of emerging work arrangements and various pillars of hybrid work arrangements, with themes for specific lessons learned under each pillar.

As is common in qualitative research [51], independent coding was performed by another researcher. The researcher was asked to identify macro and micro lessons learned (with micro lessons embedded within macro lessons), and expectations for hybrid work moving forward. To do so, a close reading of all interview transcripts was performed, followed by a second close reading which generated micro and macro lessons, which were then revised to provide a coherent understanding of the lessons.

Typically, the purpose of independent coding is to generate replicability and reliability [51,52], which allows for numeric counts of, e.g., the proportion of respondents mentioning a theme. That was not the purpose here. Instead, we sought to ensure that any theme that one coder had either missed or deemphasized, was brought to light. Specifically, instead of seeking to identify themes that occurred most frequently across, e.g., industries, any prominent theme or lesson in any single interview is valued and discussed because it may apply to others in similar circumstances.

The key theme areas emerging from the independent coding of the interview responses were utilised to form a framework for an additional visualisation step. Separate theme areas were created on an online collaborative whiteboard for each of the researchers, using the online visual collaboration software Miro, which enables distributed teams to work together synchronously in a virtual environment.

The researchers first worked independently, to re-examine the interview transcripts and field notes, in an attempt to identify potential sub-themes relating to the key areas that emerged during the coding. If a sub-theme was identified, it was added to the individual researcher's whiteboard, using a virtual sticky note. Then, once all the researchers had completed this step, the notes were merged together onto a single whiteboard, for a group discussion. The outcome of this process will be discussed in detail in the next section.

## 4. Results

Based on the independent coding steps described in the Materials and Methods section, the following table was developed to illustrate the lessons that were learned, and the frequency with which they occurred (See Table 2). From the lessons, the authors were then able to categorize a number of key theme areas they saw emerging.

**Table 2.** Key lessons and themes.

| Lessons | Appear in Interviews | Themes |
|---|---|---|
| 1. Hybrid work requires purposeful and flexible policies around when and where work is performed, accounting for customer expectations, individual preferences and circumstances, and technological resources | "purposive" 1, 4, 6, 7, 11, 12, 14 | Operations, communication, culture, technology |
| 1a. Job differences, manage divide | 1, 2, 9, 11, 12, 13, 14, 15 | |
| 1b. Customer expectations | 5, 9, 13, 15 | |
| 1c. Individual preferences | 9, 15 | |
| 1ci. Hard to get people back to workplace | 2, 11 | |
| 1cii. Survey/consult employees on preferences | 2, 4, 5, 9, 10, 11, 12, 13, 14 | |
| 1d. Individual circumstances (e.g., children, domestic viol.) | 3, 6, 9, 13, 15 | |
| 1e. Need right technological resources | 1, 2, 4, 5, 6, 7, 9, 10, 12, 13, 14, 15 | |
| 1ei. Need to use technology wisely | 1, 2, 3, 4, 5, 6, 7, 9, 10, 12, 13, 14, 15 | |
| 1eii. Hybrid meetings don't work | 3, 4, 5, 7, 10, 13 | |
| 1eiii. Importance of cybersecurity | 2, 6, 7 | |
| 1f. Tasks requiring personal interaction bad on Zoom, worse on phone | 1, 4, 7, 11, 12, 13, 15 | |
| 1g. Heightens importance of cross-function coordination | 2, 3, 5, 11, 13, 14, 15 | |
| 1h. Hard for new employees to build connections | 4, 5, 7, 11, 12 | |
| 2. Trust or clear measures of output are required for hybrid work | 1, 2, 3, 4, 6, 7, 9, 12, 14, 15 | Operations, culture, communication |
| 2a. Presenteeism undercuts hybrid work | 3, 6, 7, 9, 11, 14 | |
| 2b. Managers who prefer control undercut hybrid work | 7, 9, 13, 14 | |
| 3. Need clear policies/resource provision for home offices | 1, 3, 4, 5, 6, 9, 12, 13 | Operations, communication, infrastructure |
| 3a. Except for small orgs | 15 | |
| 4. Hybrid work easier for young people because tech savvy and not set in ways | 3, 6, 9, 14, 15 | Technology |
| 5. Walk the talk to make hybrid work | 14 | Culture |
| 6. Workplace for collaboration, as needed | 4, 5, 12, 14 | Infrastructure |
| 7. Work-from-home isolating | 1, 2, 3, 5, 7, 9, 10, 13 | Wellbeing |
| 7a. Need to plan all-in meetings, social time/events | 2, 3, 5, 6, 7, 9, 10, 11, 12 | |
| 7b. Need to plan 1-on-1s or teams to stay in touch w. employees | 2, 4, 5, 6, 10, 12, 13 | |
| 8. Mental health response important | 1, 2, 3, 5, 6, 7, 9, 10, 11, 12, 13 | Wellbeing |
| 9. Decentralize flex to team level | 2, 4, 5, 7, 9, 10, 12 | Operations |
| 9a. Adjusted work practices so recruiting improved | 7, 9, 10, 11, 12 | Future skills |
| 10. Losing employees during lockdown | 11, 12, 13 | Future skills |
| 11. Problem of excess space with partial return (hotdesk) | 3, 4, 5, 6, 7, 9, 10, 12, 13 | Infrastructure |
| 11a. Safety/loneliness issue of working alone or w. few in office | 7, 9 | |
| 11b. Flex not even for mothers prior | 3, 11, 13 | |
| 12. Pre-pandemic little flex, mainly for moms | 9, 12, 14, 15 | Operations, culture |

The key observation from this independent coding exercise was that the lessons learned from the interviews were concentrated around a small number of key theme areas: *Operations*, *Wellbeing*, *Culture*, *Communication*, *Technology*, *Infrastructure*, and *Future Skills*.

These findings will next be analysed and discussed in greater detail, to support the development of five key contributions from this research. The first is the identification of the most common hybrid work arrangements that are currently emerging, and the different degrees of workplace flexibility they offer to employees; the second is the development of a model which illustrates the key supporting pillars required for successful hybrid work; the third discusses some of the ICT infrastructure that is required to support successful hybrid work arrangements; the fourth being the theoretic contribution this research makes to the existing body of literature; before finally interpreting the results through COR theory.

## 5. Discussion

### 5.1. Emerging Work Arrangements

Five key types of post-pandemic work arrangements emerged from the discussions with participants. These included a full-scale return to the office, a move to fully remote, and three different forms of hybrid work, each offering employees different levels of location flexibility (see Figure 2). These five distinct working arrangements will now be described in greater detail, supported by direct quotes taken from the interviews.

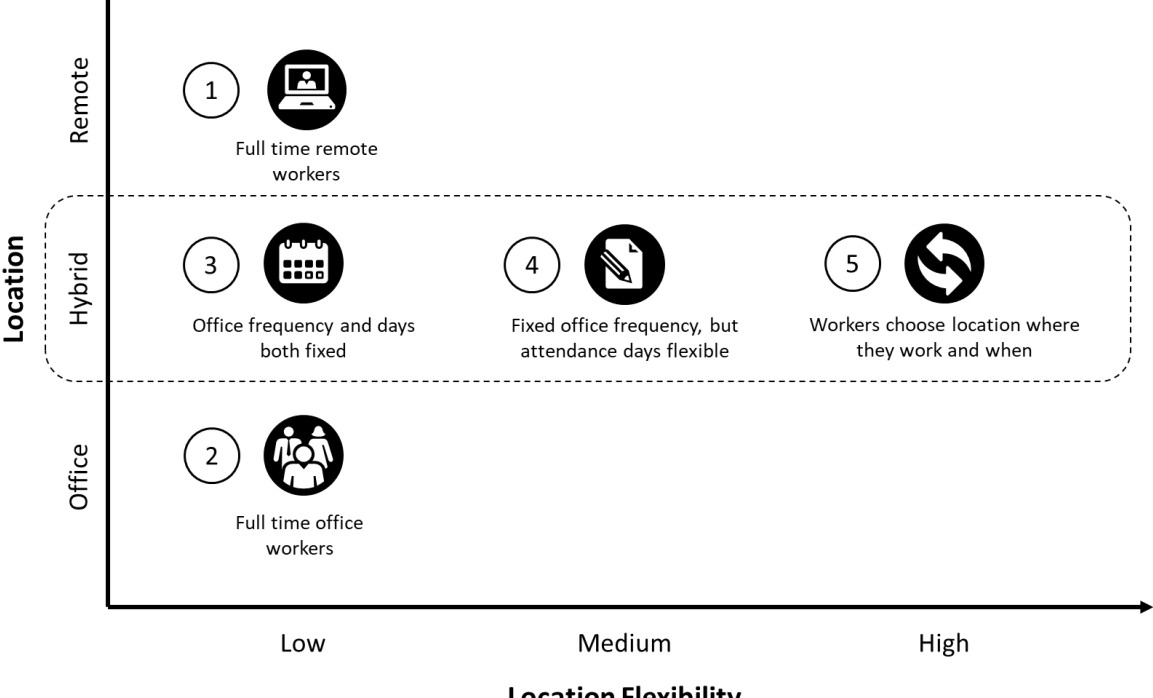

**Figure 2.** Emerging work arrangements.

1. Full-time office workers

For this arrangement, employees will find themselves working from the office on a full-time basis, as they did in the past. The key disadvantages of this arrangement are a lack of flexibility, no opportunities for dematerialisation, and the potentially negative impact on staff who may have become accustomed to flexibility in their work that was afforded during the pandemic. One key advantage is the consistency it offers, with everybody working under the same conditions, which should negate the possibility of any proximity bias occurring (i.e., managers favouring on-site as opposed to remote employees).

> "I started my life as a manager thinking people can work from anywhere at any time, and be really flexible, and I very quickly stopped that because I found that our teams were drifting apart . . . going through COVID has affirmed my view that people have to work together . . . fireside chats that you have around the water cooler or having a coffee means something to people". (A1)

2. Full-time remote workers

Here, employees must work remotely all the time. One of the organisations we interviewed during this research (K1), had transitioned to this arrangement during the pandemic and planned to operate as a fully remote business going forward, whilst others were considering it. This arrangement offers significant opportunity for dematerialisation, access to a wider talent pool, and minimises the chances of proximity bias occurring. However, location flexibility is low, and workers probably do not have the opportunity to

formally meet with colleagues and collaborate in the physical world as regularly as they would with other work arrangements.

*"So, our approach now is that no more than 20 per cent of the workforce will return to the office and we are actively recruiting people who will never be in the office". (K1)*

*When asked about WFH during COVID—"I think what they're seeing, what leaders are seeing, is that it's really not impacting engagement, it's not really impacting productivity and so if anything, it's kind of helping engagement, people are more satisfied. So really I think this way of working is really going to be part of our ongoing employee value proposition as well". (Q2)*

3. Office frequency and days both fixed

This was the first of the hybrid arrangements we encountered and requires employees to attend the office on a mandated number of fixed days per week. In the case of N1, this was Tuesdays, Wednesdays, and Thursdays, with everybody WFH Mondays and Fridays.

This arrangement is straightforward to manage, as everyone knows where everybody else is on any particular day, and proximity bias is minimised. However, there is no flexibility to cater for different office day preferences, and no dematerialisation is possible, as the office still needs to accommodate its full quota of workers three days per week.

*"You must be in the office two days a week. Wouldn't want everyone in at the same time. And if you're customer service facing, you know that might be maybe more frequent. We would aim to try and have maybe the office closed Friday and Monday and have it open Tuesday, Wednesday, Thursday . . . or whatever works best for the business, so that we get a reasonable population to get that social connectivity back". (H1)*

*"Come in on a Monday and a Thursday" (M1)*

4. Fixed office frequency, but attendance days flexible

With this arrangement, it is mandated that workers attend the office for a specified number of days (e.g., 3 days per week, or 5 days per fortnight, etc.), but employees are given the flexibility to choose which ones. This arrangement offers employees greater location flexibility than Arrangements 1–3, but requires greater coordination to manage, and could lead to instances of proximity bias. Over time, based on office occupancy data, this arrangement could support some dematerialisation of infrastructure.

*"Two days of your choice, it would seem at this stage, is the way that's going to be interpreted". (I1)*

5. Full flex—workers choose the location where they work and when

The final hybrid arrangement we observed was one offering workers complete choice over when they come to the office full time, if at all, and when they work remotely. This obviously offers employees the greatest level of flexibility possible, but appears to be the most difficult to coordinate, has the least consistency, and is the most likely to result in proximity bias. Similar to Arrangement 4, this arrangement could also result in a dematerialisation of infrastructure, over time.

*"Rather than, 'you must be in the office a minimum of two or three days a week', which never was the mandate or the case here, the focus is going to continue to be on, come together purposefully and meaningfully with your team to interact and collaborate face to face with your team". (N1)*

*"(I think) it would be nice for people to be there one to two days but there's nothing mandated". (C2)*

We also observed variations of these arrangements, which combine certain elements, from two or more of the models. These include a set number of days in the office, some days are fixed but others are flexible—e.g., five days in the office per fortnight (one is fixed, the other four are chosen by the employee)–

*"The system in place is for, like, 5 days per fortnight and everybody in on one set date. We're suggested that they have one common day per fortnight where everybody is on site from their team, and they can actually use that one day for workshops and meetings". (L1)*

Similarly, some organisations were considering a full-time remote working model apart from one day in the office every 3 months, and mandates a maximum number of days in the office per week (e.g., no more than three), instead of a minimum number.

Some organisations also adopted multiple arrangements, across their different business units, such as the food grower who allowed their administration workers to work in a hybrid arrangement, while their farm labourers had to be full-time on-site. Others coordinated Arrangement 3 so that different departments worked from the office on different days, in an attempt to maintain consistent attendance numbers each day.

*" . . . the discontent created between the right and the privilege of working from home and having to come to work, that's a very sensitive topic in the organisation at the moment". (A1)*

### 5.2. Five Pillars of Successful Hybrid Work

Using the key lessons and themes emerging from the interviews, the authors were able to develop the following model, which we call our *five pillars for successful hybrid work* (see Figure 3).

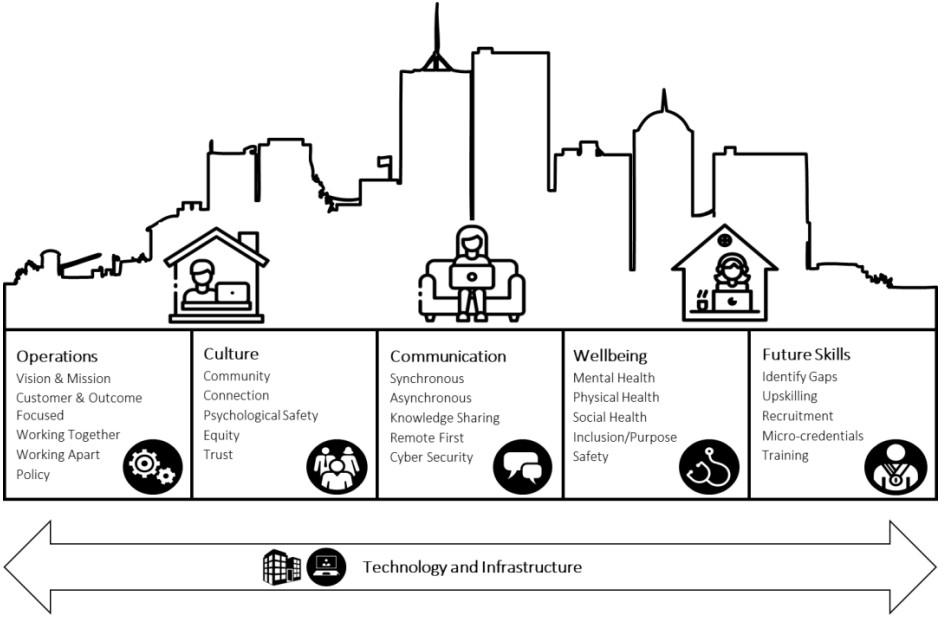

**Figure 3.** Five pillars for successful hybrid work.

These five pillars for successful hybrid work will now be discussed in detail.

1. Operations

It was clear from the interviews that, whatever work arrangements were going to be adopted, they needed to align with and support the existing focus, vision, and mission of the organisation. The key focus of any organisation is to deliver value to customers, so that had to be maintained or improved upon.

A number of the participants talked about the office as being a 'place of purpose', and that it would be difficult to attract workers into an office to simply conduct tasks they have been comfortably performing at home for nearly two years.

*"I know there are some offices that have gone with the two days at home, three days, whatever it might be, we've also shied away from that. Our key thing is if you're going in, you're going in for a reason . . . the idea is that the office is a place of purpose, so if you're going in, you're going for a reason, you're not going in because it's a Monday". (Q2)*

Examples of 'purpose' for the office, include collaboration, community building, and a place for connecting with colleagues. Some work may be performed better in the office, whilst other tasks might work better remotely, so it is important to consider the nature of typical work tasks when selecting the best hybrid arrangement for a particular team, department, or organisation.

Thirteen participants indicated that their organisations had a flexible work policy prior to the pandemic, but all of them said that it had now been updated, and their current definition of workplace flexibility was very different to what it was pre-pandemic.

*"I think prior to COVID working flexibly meant something very different and not because we said you can't do certain things, but because people never really thought about it". (J1)*

*"There might have been some ad hoc work from home if someone had a plumber coming that day, and they could log in remotely, but there was nothing formalized. In fact, it was probably actively discouraged". (L1)*

The decisions regarding what work arrangement would be implemented were commonly made by a multi-disciplinary working group or task force, but the recommendations they made were typically flexible enough to accommodate line manager discretion and support operational requirements at the team/department level. This promotes increased autonomy for employees at all levels, as opposed to the "elitist phenomenon" previously described by Pyöriä (2011), in alignment with SDG8.

2. Culture

Another theme arising from the semi-structured interviews was the importance of establishing and maintaining an aligned workplace culture in a hybrid setting.

*"We want to build an inclusive working environment and culture and that sense of belonging. It's critical. It's crucial to inclusion . . . I think that's one for us to be really mindful of, and to work towards, in this new style of working". (N1)*

*"Our culture has changed, and I wouldn't say that it's bad, but I wouldn't say that we actually have our finger on what it is right now and we need to understand how do we continue to have that collaboration and that connectivity with our people in an environment that we're now operating in and will continue to operate in for a very long time". (L1)*

A number of ideas emerged to support culture building in a hybrid working environment, including holding a pitch day, running a challenge to raise money for a cause together, virtual wine tasting and coffee roulette, which randomly matches colleagues for a virtual coffee chat. One common theme for nurturing workplace culture was to give teams ownership of their ideas to encourage culture building from the bottom up.

3. Communication

The ways in which organisations communicate in a hybrid work arrangement, having some employees at the office and others working remotely at any one time, requires different tools and processes. The importance of videoconferencing software was frequently emphasised, in addition to online collaboration tools, such as Miro and Mural, and asynchronous communication platforms such as Slack.

*"We switched over to being a full Microsoft Office shop in order to have Teams and we use a number of the functions that come with that Microsoft Office. So, there's planner boards to plan our work. Obviously, Teams to facilitate the work and some other things will use outside of that product suite of things like jam boards and Mural boards for collaboration and working together . . . if you'd spoken to me in February 2020, I would have said to you, we don't know what these tools are". (L1)*

*"We use it (Slack) for everything, and I think one thing actually that has changed is I don't send as many emails anymore. I have slack groups and I send it in Slack and it makes the conversation between people much easier". (J1)*

*"(for meetings) if one person's on a Zoom call, everyone should be on a Zoom call". (Q2)*

*"None of us had teams up until about three weeks until into the pandemic, we were all operating off email for the most part". (I1)*

When teams are distributed, such as they are in a hybrid or remote work arrangement, it was underlined that cyber security also becomes an increasing concern.

*"We will have no desk phones. Everyone's on mobile phones. We are introducing a new voice over IP system via Teams. We have lots of cyber security rules so we need to cross a few hurdles before we can implement that". (C1)*

*"I think one of the things would be, from a security perspective, can we do BYO devices, what does that look like and that sort of thing. But again, that's being explored at the moment". (Q2)*

4. Wellbeing

One of the most prominent themes to emerge from our interviews, was the increased focus on the health and wellbeing of staff, in alignment with SDG3 targets.

*"I worked to establish a rhythm in weekly manager employee meetings where if you were discussing nothing else, you were at least discussing wellbeing. So, I established this rhythm that every manager was asking their employee about wellbeing". (N1)*

*"We've done lots of work around providing mental health support, we've created our wellbeing ambassador team who are actually employees that are trained. They do mental health first aid training, and they are a point of contact for people who want to reach out to them, in addition to our employee assistance program". (L1)*

*"We did the duty of care for working from home based on case law. Fatigue management, that sort of thing . . . How do I cope? How do I change? How do I adopt? How do I thrive?" (H1)*

In addition to regular check-ins with their managers, other ideas designed to support employee wellbeing included virtual exercise classes or challenges (e.g., daily step count), virtual team lunches and coffee breaks, online quizzes, book clubs, and movie nights. Additional time off was another theme that emerged, particularly for teams who were still in lockdown:

*"We have 'Wellbeing Wednesdays' every Wednesday . . . lunchtime sessions with psychologists and mindfulness coaches . . . (and) from July through to September, and now from October through to December, we have given everybody a day off a month to just reset so they don't have to access their annual leave. So that's six days in six months, one day a month". (N1)*

Having ergonomically compliant home office setups, and access to all the tools needed to perform work roles across both locations is also a requirement for successful hybrid work arrangements.

*"One is a virtual home/office audit and there's a safety checklist that both the manager and the employee have to sign. It's not just a post and pray, tick off. It's a virtual audit". (C1)*

*"We already had some work that had been done around ergonomic setups for home, but that actually got formalized and made more broad, so we had a policy for that in a process and we've got checkpoints around. That's digital photos, risk assessments etc., that employers need to do". (L1)*

5. Future Skills

The talent and skills requirements for operating as a hybrid workforce are predicted to be different than those of a traditional face-to-face office environment. In particular, the roles and responsibilities of line managers/supervisors have changed, and these new ways of working require different leadership skills. Managers now need to deal with

more emotional issues than before, requiring different communication styles, and new coordination issues with distributed teams.

> *"I've got a psychologist coming in working with our managers on a bit of a workshop. How are you there for your people? What sorts of challenges are they experiencing? How can you support them? How can you take care of yourself?" (N1)*

> *"A lot of training has been done. Both formal agile-based training and coaching from agile coaches. With that group . . . we also decided that it would be really important to do emotional intelligence". (L1)*

In addition to upskilling existing staff, the requirements of new staff are also expected to change, which impacts future recruitment strategies.

> *"It gives us a lot of opportunity to recruit from a different pool of people if we're going work from home because we can—for example, we never would have recruited people from Western Australia (-3hrs from Australian Eastern Daylight Time) but now we're actively pursuing that because they can cover the late shifts and not have to pay penalty rates because it's not late for them". (K1)*

When transitioning to different ways of working, it is inevitable that organisations will have to develop or train new skills internally, or recruit new skills externally, to better support the new direction of the operation.

*5.3. ICT Supports for Hybrid Work*

Underpinning these five pillars for successful hybrid working is a requirement for appropriate technology and infrastructure, that is fit for purpose for this new way of working. For instance, when employees first started working from home, at the start of the pandemic, many organisations had to purchase laptops, monitors, various software applications, etc., to support this sudden transition.

> *"So, we went from having a small section of people that were working remotely to then everyone receiving VPN access virtually overnight and up to 4000 employees went online virtually. At that time, we also introduced our working from home package, which was to provide all the resources needed, hardware, software, furniture and the like, so people could have that home office setup at the expense of the organisation". (M1)*

Long before the pandemic, digital technologies had already evolved to the stage where they facilitated a large proportion of work tasks being location-independent. Many knowledge workers continued to work in offices full time, not because they needed to be there to do their jobs, but because of the workplace norms established over the decades leading up to that point. COVID-19 disrupted these norms, resulting in many more workers now accessing FWAs such as WFH, in the form of the hybrid work arrangements that are popular today.

It is clear from the findings of this research, and this rise in the popularity of hybrid work arrangements, that ICT will play an increasingly important role in facilitating and supporting this practice. In this section, we discuss a range of examples of ICT solutions, which the researchers have identified as playing a critical role in supporting the pillars for successful hybrid work.

5.3.1. Operations

One of the key operational challenges of hybrid work arrangements is managing employees who split their time between different locations and providing them with the appropriate support to enable them to do that effectively.

Desk booking apps such as Officely, Kadence, and Envoy, sometimes called *hoteling* or *office hoteling* apps, are software tools that enable workers to reserve desks, workstations, or meeting spaces ahead of time, for the periods they will be spending at the office. They enable employees to see who else will be on-site, and where in the office they will be sitting,

and allow employers to know in advance how many people will be working on-site on any particular day, which helps with managing both capacity and resources.

Another key challenge operational challenge of hybrid work arrangements is workflow management. Participants indicated tools such as Monday and Trello play an important role, in enabling distributed teams to organise their work activities effectively, whilst tracking the progress of work tasks and important milestones. Cloud-based file management systems such as Google docs and Microsoft OneDrive also enable documents and knowledge to be shared amongst distributed teams, allowing multiple people to work on them at any one time, regardless of location.

### 5.3.2. Culture

Another major challenge facing hybrid workforces is how to nurture trust, connection and community, and nurture a culture of psychological safety when employees are distributed and regularly switching work locations. Tools such as Fond and Nectar help to strengthen team culture and build morale, by enabling employees to publicly acknowledge their colleagues' achievements and award digital gift cards in recognition of good work, etc., via an online platform. Similarly, we found participants were using ICT tools such as Culture Amp and Everperform to quickly collect and analyse employee data, via mobile-enabled pulse surveys, to monitor their organisational culture.

As they continue to mature, new products such as Meta Horizon Workrooms, NVIDIA Omniverse and Microsoft Mesh, will enable workers to build relationships with colleagues and teammates in the virtual world, in a more sophisticated way than most contemporary technologies allow.

### 5.3.3. Communication

Effective communication technologies are critical for successful hybrid work and range from web-conferencing systems (WCS) systems such as Zoom and Teams to online messaging platforms such as Slack, and even simple email. WCS technologies played a key role in the pandemic transition to WFH [53], and will remain an essential part of hybrid work models.

These technologies enable knowledge workers to communicate with their colleagues, customers, suppliers, and partners, all around the world, both synchronously and asynchronously, and there is a wide range of alternatives available to match the needs of organisations of different sizes and budgets. Online collaboration tools such as Miro and Mural take this a step further, enabling distributed teams to work together in an online environment, for visual collaboration activities such as brainstorming.

In a similar manner to how technologies will evolve to better support hybrid work culture, improved VR and Metaverse environments are also likely to offer more immersive experiences for virtual communication and collaboration.

### 5.3.4. Wellbeing

Employee wellbeing became a key focus during the pandemic and continues to be a priority for hybrid work arrangements, and ICT tools specifically designed to monitor mental, physical, and social health, are growing in popularity.

OKPulse uses surveys, analytics and artificial intelligence to assess employee wellbeing against a range of health and wellness benchmarks, as an early detection mechanism for signals of stress and anxiety. Whilst another Australian app, Headspace, offers hundreds of meditation and exercise programs for improving employee stress, focus, sleep, and movement.

### 5.4. Theoretical Contribution

The identification of these post-pandemic hybrid work models, and the key pillars required to successfully support hybrid work arrangements, make a valuable and timely contribution to this newly-emerging field.

The academic literature describing the different forms of hybrid work models that are being adopted is lacking, and most studies in this field do not go beyond describing the balance of the number of days in the office, versus days spent working remotely. This paper gives adds some much-needed granularity, by describing a number of practical features and considerations for hybrid work missing from the existing literature, such as parameters around remote work frequency and mandated attendance days, that control different levels of employee location flexibility.

Academic studies investigating the typical supports needed for hybrid and remote work, with theoretical and empirical analysis of the conditions under which hybrid work might be successful, are starting to emerge. For example, Burleson, Eggler and Major [54] recently examined a range of diversity, equity, and inclusion (DE&I) challenges facing remote workers; whilst Knight et al., [55] discuss the benefits of hybrid work vs fully remote work from a loneliness perspective; and Odum, Franczak and McAllister [56] allude to the risk of proximity bias, underlining the importance of ensuring equity between both in-person and remote employees for nurturing successful hybrid work arrangements. However, studies like these have a much narrower focus and no current literature identifies the range of different pillars needed to support successful hybrid work models, in the same way, this paper does.

Similarly, Chong et al. [36] and Wang et al. [40] studied a range of emotional stressors and wellbeing challenges, experienced by those forced to work from home in the early stage of the pandemic. However, whilst these challenges are an important consideration for future work models, post-pandemic hybrid work arrangements were not the focus for these authors.

Moreover, whilst the likes of Gajendran et al. [37] investigated job performance and social concerns associated with telework many years before the pandemic, and Gilson et al. [38] conducted a 10-year review of the academic literature discussing virtual teams, existing knowledge on remote working like this is now being questioned in the wake of the pandemic [40], with the parameters around adoption and attitudes to different modes of working having changed so significantly since COVID-19. It is important to remember at this point that there is a distinct difference between virtual teams (or remote workers), and hybrid ways of working—hybrid workers are more likely to spend time in the office with colleagues on a regular basis and may therefore not face the same challenges as those who work remotely for longer periods.

Equally, it is crucial to extend the post-pandemic academic literature to address traditional telework challenges and negative impacts, using a contemporary hybrid work lens. This provides a practical contribution for practitioners, and a theoretical contribution for future research, by identifying ways in which today's hybrid leaders are tackling concerns such as rising work intensification due to increased flexible work [39], altered physical and temporal boundaries [57], and work–life balance [58].

In the non-academic literature, Gratton [59] produced a widely-respected visualization of hybrid work, based on the axes of time and place. With reference to these axes, Gratton [59] encourages managers to think about how jobs, tasks, projects and workflows might be affected by changes in the location and timing of work, to understand employee preferences when designing hybrid work, paying particular attention to inclusion and fairness. These are valuable contributions to the wider discussion, but no mention is made of what different types of hybrid work models are emerging, or other important considerations such as wellbeing or hybrid work culture, etc.

Overall, the authors believe that this paper makes a very significant contribution, to a high-priority field of study.

### 5.5. Hybrid Work through a COR Lens

The earlier discussion of COR theory suggested that forced WFH arrangements enhanced important resources for employees, including increased autonomy, flexibility, and reduced commuting time, with managers and supervisors losing a key resource in terms of

control and requiring new resource investments in terms of coordination and performance measurement. Employees preferred some type of hybrid arrangement in the future to provide valuable social resources but did not wish to lose all that they had gained from WFH in the process.

At this point, COR theory could in this case be effectively reduced to a tautology: WFH during the pandemic altered resources and work practices such that hybrid work arrangements were thereafter superior for many organisations. For example, several cases are provided above of organisations that invested in capabilities for virtual meetings and WFH. Given those investments, and resulting levels of WFH performance, hybrid work arrangements were superior thereafter.

Instead, we interpret the results in terms of net resource gains for employees which they strove to maintain, at least in part, through hybrid work arrangements. Three pieces of evidence support this view, that have significant practical implications for managers, which will be discussed in detail in the following section.

### 5.6. Key Managerial Implications

Firstly, with few exceptions, managers believed their employees wished to engage in hybrid work arrangements instead of full-time on-site work. As one put it, "the majority would like to work in a blended type arrangement" (Q1), which is precisely the plan moving forward. Another stated, "there's not a big appetite to be in the office [full-time]" (K1), in the case of an organisation currently operating full-time remotely, with plans to continue this arrangement in the future. Yet another in a fully flexible workplace stated, "I think everyone quite likes an element of flexibility and working from home and being able to walk the dog and all that sort of stuff". (C2). One manager in agriculture had 75% of employees on-site throughout lockdown, with 25% WFH, and did not understand why the 25% were having "a hard time coming back to work", (A1) implying that he did not grasp the resource gains experienced by those employees.

Second, there were some preferences for a return to full-time on-site work among managers. For example, the same manager who believed the organisation would end up mandating at least three days on-site per week also noted that, "from a management perspective, we would like staff to be on site" (Q1). Even in a workplace that planned for full-time remote work into the future, "we've had to do a lot of work on trust" among supervisors (K1). In one case, a planned return to full-time on-site work was because "my position would be to get everyone back in the office" (A1), explicitly stating that this is the preference of management. Yet another noted, "we lost some candidates the other week because they were in New Zealand and the general manager really wanted to meet them face-to-face and couldn't" (C2). A manager who planned on varied types of hybrid work in the future also mentioned an employee who answered work calls with the television on in the background. In response, "when we open up, she will be back for her nine days because otherwise you're stupid". (M2). In a case where the manager believed future arrangements would involve two days on-site per week chosen by the employee, it was also mentioned that:

> So, the folks who do two days they don't get the promotions. The ones that do three do and then all of a sudden everyone look for [five days] … and then we'll be back where we started. There's a very good chance of that happening. (I1)

In other words, this manager believed other managers would reward on-site work, eventually driving all employees back to full-time on-site arrangements.

Third, hybrid work requires greater coordination of resource expenditures from managers relative to either full-time remote or full-time on-site. A close reading of the five pillars described above supports this view, as operations require making the workplace a 'place of purpose', with a constant juggling of tasks that can or cannot be performed off-site, the resources needed to establish a culture with a 'sense of belonging', given many employees are often remote, resources for communication and how those are used for partly on- and partly off-site employees, employee wellbeing in this new environment,

and particularly new requirements of managers in terms of coordination, trust, and performance management. One manager we mentioned earlier as noting that employees would be expected to be on-site five days per fortnight, went on to tie together managerial preferences for full-time on-site work and coordination resources:

> *But the coordination of trying to organise things with a group of people where you may only have half of them in the office at any one time, they found logistically a little bit challenging, and would prefer that we didn't actually have people doing five days a fortnight out of the office. (L2)*

It is reasonable to conclude that for many managers, hybrid work arrangements involve managerial resource expenditures beyond those required for either full-time remote or full-time on-site arrangements. As a result, it is reasonable to conclude that it is the context of employees experiencing resource gains under WFH that drove their efforts to promote hybrid work. An implication is that some managers may, over time, force movement back to full-time on-site work, thereby reducing managerial resource expenditures.

## 6. Conclusions

The COVID-19 period acted as a catalyst for greater understanding and uptake of flexible work arrangements, the mass adoption of telecommuting practices identified 50 years ago [23–25], and the emergence of new 'hybrid' work models as the dominant arrangement for knowledge workers. This period of forced WFH participation disrupted the established status quo of work arrangements. More importantly, it has led to overcoming many of the pre-pandemic barriers identified by Hopkins and McKay [22].

In the wake of the pandemic, organisations are adopting a range of different hybrid work models, which attempt to combine the benefits of mixing office work and working remotely, and have the potential to support better working conditions (SDG8) and improved wellbeing outcomes (SDG3). This makes it necessary to update our previous understanding of these practices and frame this current manifestation of hybrid work within a contemporary context, whilst extending the traditional academic literature from this field. The authors have attempted to do this, by revisiting the key terms and definitions with a post-pandemic lens, to position hybrid work within an academic context (see Figure 1).

This research offers unique insights into the format of the hybrid work models which are now emerging, and how these new arrangements are being designed and implemented in practice, based on the experiences of senior HR professionals. The findings offer significant value to academic researchers and industry practitioners alike, by not only identifying the most popular hybrid work models being adopted, but also the key components required to successfully support these arrangements. To successfully support hybrid work arrangements, managers must take into consideration the operation, workplace culture, communication methods, wellbeing of staff, and address any new skills gaps that may have emerged due to the change in work protocols.

ICT will play a major role in providing and sustaining this support, through the likes of wellbeing tools, desk booking apps, workflow management software, and applications for strengthening team culture. WCS technologies were a vital ingredient in the transition to WFH during the pandemic and will continue to be an important part of hybrid communication, as will online collaboration tools such as Mural and Miro. As technologies such as VR and Metaverse evolve, they are expected to provide increasingly sophisticated environments for online interaction and collaboration, negating the need for face-to-face in-person contact even further.

Finally, the COR analysis of the results suggested that employees who experienced WFH during the pandemic were the primary driver for hybrid work arrangements, with at least some managers preferring a return to full-time on-site arrangements. Anecdotal evidence supports this possibility. Goldman Sachs announced a return to full-time on-site in March of 2022 [60]. However, by May of that year, companies that clearly preferred full-time on-site, such as JPMorgan Chase and others in banking, finance, tech, and real estate, had each given in to the demands of employees for hybrid work, backed up by tight

labour markets [61]. How these dynamics will continue to play out over time remains a question for further research.

Whilst the authors have great confidence in the significance of this research, and the range of benefits and value it offers to both academics and HR practitioners, it is important to acknowledge that there were also several limitations to the study. Our interviews only captured the opinions of Australian practitioners and, whilst there will no doubt be many similarities in the way hybrid models have been designed and managed in other parts of the world, it is likely there will be regional differences, too. Similarly, although our research was designed to investigate the different hybrid models emerging across a range of different industry sectors, several sectors were not represented in the study, and it is possible that they may yield alternative results. Finally, as our interviews were conducted solely with managers and not employees, it is important to also capture the employees' perspective in future research, to validate some of our findings and our hypothesis regarding the resource gain cycle due to working from home. These areas of limitation all pose interesting opportunities for future research.

**Author Contributions:** Conceptualization, J.H. and A.B.; Methodology, J.H. and A.B.; Formal analysis, J.H. and A.B.; Investigation, J.H. and A.B.; Writing—original draft, J.H.; Writing—review & editing, J.H. and A.B.; Project administration, J.H. All authors have read and agreed to the published version of the manuscript.

**Funding:** This research received no external funding.

**Institutional Review Board Statement:** This project was approved by or on behalf of Swinburne University of Technology's Human Research Ethics Committee (SUHREC) in line with the Australian *National Statement on Ethical Conduct in Human Research*.

**Informed Consent Statement:** Informed consent was obtained from all subjects involved in this study.

**Data Availability Statement:** Whilst there is no formal intention for the data (or samples) from this investigation to be made publically available, for future research projects by other researchers, the named researchers welcome such requests and will respond to them on a case-by-case basis.

**Conflicts of Interest:** The authors declare no conflict of interest.

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
