# Peer review of "The Future Is Hybrid: How Organisations Are Designing and Supporting Sustainable Hybrid Work Models in Post-Pandemic Australia"

_sustainability, doi:10.3390/su15043086_

Round 1
Reviewer 1 Report
This is a very interesting and timely topic where the authors investigate for a better understanding of the hybrid working environment and extend previous academic understanding of this practice, using a post-pandemic lens.
Please find my comments here:
- Abstract is written very well.
- Introduction section is also in a good shape. I would suggest expanding this section.
- Analysis/Results and Discussion looks good and very well written.
- All the figures are very well developed. Figures 2 and 3 are a bit blurred.
- Theoretical contribution is great, but missing the managerial implications section. Some of it is covered under "Hybrid work through a COR lens", but it is good to have a separate section for managerial implications.
- Conclusion section is also written very well. Having said that, please elaborate more on the limitation of this study.
To conclude, the current clarity, content and flow of the manuscript are very strong. The submitted manuscript needs some minor improvement to satisfy the expected level for publication.
In addition, I would highly suggest the authors review as many as possible published articles by the targeted journal and consider citing those relevant to this study.
All the best and good luck!
Author Response
Dear Reviewer 1, please see our response to your invaluable feedback comments, in the document attached.
Best regards,
John

Reviewer 2 Report
Thank you for the opportunity to review the paper “The future is hybrid: how organizations are designing and supporting sustainable hybrid work models in post-pandemic Australia”. It is a very well drafted and original research proposal offering a new and well-conceptualized model for successful hybrid working, and significant insights into post-pandemic work arrangements in Australia. I have following recommendations for the authors:
§ Minor revision of a few sentences is required in terms of grammar and sentence structure. (Line # 251, 463, 539, 582, and 607)
§ Under the heading ‘Third Places’, the authors stated “In urban planning, third places are viewed as an important ingredient in community building, and for stabilizing neighborhoods. They define them as social surroundings, separate from the two usual social environments of home - "the first place," and work - "the second place". Please add a reference for the definition given in line # 100.
§ Under heading 2.2. Conservation of Resources Theory, it is mentioned that “Although originally developed as a theory of stress (Hobfoll, 1989), it has since been expanded to include a wide variety of adverse circumstances, ranging from natural disasters to sex trafficking, and various workplace issues (Giddens, Petter, & Fullilove, 2021; Holmgreen, Tirone, Gerhart, & Hobfoll, 2017)”. What is the reasoning of adding these line in lieu of COR.? Please rewrite what you are trying to explain here.
§ The theoretical and practical implications of the study are well-written. The authors should also add the limitations of the study and recommendations for future research. As it is stated by the authors that “As a result, it is reasonable to conclude that it is the context of employees experiencing resource gains under WFH that drove their efforts to promote hybrid work”. The conclusion about employees experiencing resource gain cycle due to working from home is reasonable and supported by the COR theory. But the interviews were done with managers and not employees, it should be added as a limitation of the study.

Author Response
Dear Reviewer 2, please see our response to your invaluable feedback comments, in the document attached.
Best regards,
John
